Expression profiles of circular RNAs and interaction networks of competing endogenous RNAs in neurogenic bladder of rats following suprasacral spinal cord injury

Ruan Jimeng
Cui Xin
Yan Hao
Jia Chunsong
Ou Tongwen outongwen1967@126.com
Shang Zhenhua shangzhenhua16@126.com
Department of Urology, Xuanwu Hospital Capital Medical University , Beijing , China
Uversky Vladimir
Electronic publication date: 2023 Sep 18
Publication date: 2023
Volume: 11
Electronic Location ID: e16042
Received 2023 May 30; Accepted 2023 Aug 15
Copyright: ©2023 Ruan et al.
Copyright year: 2023
Copyright holder: Ruan et al.
License: This is an open access article distributed under the terms of the Creative Commons Attribution License, which permits unrestricted use, distribution, reproduction and adaptation in any medium and for any purpose provided that it is properly attributed. For attribution, the original author(s), title, publication source (PeerJ) and either DOI or URL of the article must be cited.
License URL: https://creativecommons.org/licenses/by/4.0/

Keywords: Suprasacral spinal cord injury, Neurogenic bladder, Bladder fibrosis, circRNAs, Transcriptome high-throughput sequencing, Co-expression network, Biomarkers

Funding: National Natural Science Foundation of China 82100819 Capital Health Research and Development of Special Fund 2020-2-2015 Beijing Hospitals Authority Youth Program QML20230808 Training Fund for Open Projects at Clinical Institutes and Departments of Capital Medical University CCMU2023ZKYXY019 This work was supported by the National Natural Science Foundation of China (NO. 82100819), the Capital Health Research and Development of Special Fund (NO. 2020-2-2015), the Beijing Hospitals Authority Youth Program (NO. QML20230808), and the Training Fund for Open Projects at Clinical Institutes and Departments of Capital Medical University (CCMU2023ZKYXY019). The funders had no role in study design, data collection and analysis, decision to publish, or preparation of the manuscript.

==============================
Background

Neurogenic bladder (NB) following suprasacral spinal cord injury (SSCI) is an interstitial disease with the structural remodeling of bladder tissue and matrix over-deposition. Circular RNAs (circRNAs) are involved in fibrotic disease development through their post-transcriptional regulatory functions. This study aimed to use transcriptome high-throughput sequencing to investigate the process of NB and bladder fibrosis after SSCI.

Methods

Spinal cord transection at the T10–T11 level was used to construct the SSCI model in rats (10–week–old female Wistar rats, weighing 200 ± 20 g). The bladders were collected without (sham group) and with (SSCI 1–3 groups) NB status. Morphological examination was conducted to assess the extent of bladder fibrosis. Additionally, RNA sequencing was utilized to determine mRNAs and circRNAs expression patterns. The dynamic changes of differentially expressed mRNAs (DEMs) and circRNAs (DECs) in different periods of SSCI were further analyzed.

Results

Bladder weight, smooth muscle cell hypertrophy, and extracellular matrix gradually increased after SSCI. Compared with the sham group, 3,255 DEMs and 1,339 DECs, 3,449 DEMs and 1,324 DECs, 884 DEMs, and 1,151 DECs were detected in the SSCI 1–3 groups, respectively. Specifically, circRNA3621, circRNA0617, circRNA0586, and circRNA4426 were significant DECs common to SSCI 1–3 groups compared with the sham group. Moreover, Gene Ontology (GO) enrichment suggested that inflammatory and chronic inflammatory responses were the key events in NB progression following SSCI. Kyoto Encyclopedia of Genes and Genomes (KEGG) pathways enrichment associated with the “Chemokine signaling pathway”, the “IL-17 signaling pathway”, and the “TGF-beta signaling pathway” suggests their potential involvement in regulating biological processes. The circRNA–miRNA–mRNA interaction networks of DECs revealed rno-circ-2239 (micu2) as the largest node, indicating that the rno-circ-2239–miRNA–mRNA–mediated network may play a critical role in the pathogenesis of SSCI-induced NB.

Conclusions

This study offers a comprehensive outlook on the possible roles of DEMs and DECs in bladder fibrosis and NB progression following SSCI. These findings have the potential to serve as novel biomarkers and therapeutic targets.

Introduction

It is a devastating experience to suffer a suprasacral spinal cord injury (SSCI), which can result from traumatic or non-traumatic events involving a fractured or dislocated spinal column. The most common cause of the trauma is traffic accidents, followed by falls (Singh et al., 2014). In patients with SSCI, neurogenic bladder (NB) is one of the most common causes of death because it presents various complications (Fang et al., 2020). Lower urinary tract complications include detrusor-sphincter dyssynergia, detrusor overactivity, and bladder fibrosis (Kavanagh et al., 2019); and upper urinary tract complications include ureteral reflux, ureteral dilatation, pyelonephritis, hydronephrosis, which in turn progress to renal failure or even death (Groen et al., 2016). Severe acute SSSCI first severely impairs lower urinary tract function, leads to severe bladder fibrosis in a short period of time, and can significantly affect the quality of life and life expectancy of patients (Hu, Granger & Jeffery, 2016). Therefore, preventing bladder fibrosis and protecting upper urinary tract function is the key to improving NB prognosis following SSCI and improving patients’ quality of life.

RNA molecules play a complex role in organisms, with a diverse range of RNAs involved, such as messenger RNAs (mRNAs) that code for proteins and non-coding RNAs (ncRNAs). The main types of ncRNAs are microRNAs (miRNAs), long non-coding RNAs (lncRNAs), and circular RNAs (circRNAs). One of the functions of ncRNAs is to act as a competitive endogenous RNAs (ceRNAs) and are crucial for regulating post-transcriptional level gene expression (Salmena et al., 2011). CircRNAs are a group of non-coding RNAs commonly expressed in eukaryotic cells, and such endogenous RNAs are characterized by stable structure and high tissue expression specificity (Zhou et al., 2020). CircRNAs have been shown to serve as inhibitor of miRNAs, competitively inhibit miRNAs activity (Kong et al., 2020), and further reduce the levels of specific miRNAs in cells, which can be used as a treatment for disease-related miRNAs depletion (Tost, 2018). CircRNAs were implicated in fibrosis regulation in the bladder and other organs. CircPVT1 has been proven to promote bladder outlet obstruction (BOO)-induced bladder fibrosis through the down-regulation of miR-203v, and the BOO-induced initiation of the circPVT1-miR-203-SOCS3 axis further promotes extracellular matrix deposition in the bladder wall (Romo et al., 2018). The use of transcriptome high-throughput sequencing analyses to investigate how miRNAs, lncRNAs, circRNAs, and mRNAs interact in various physiological processes, such as bladder cancer and bladder fibrosis, has become widespread due to the progress in sequencing technology (Li et al., 2018b). Despite this increase in interest, circRNAs mechanism in bladder fibrosis pathogenesis, particularly NB following SSCI, is rarely reported. The study could improve our comprehension of how regulatory mechanisms work in NB following SSCI, provide critical backing for subsequent studies on bladder fibrosis, and stimulate novel theories for the pathogenesis and treatment of NB following SSCI.

The current study employed transcriptome high-throughput sequencing to detect the expression profiles of mRNAs and circRNAs in NB tissues at three different time points following SSCI and in sham group bladder tissues. The dynamic changes of differentially expressed mRNAs (DEMs) and differentially expressed circRNAs (DECs) in different time points of SSCI were further analyzed; pathological features in the progress of bladder fibrosis were identified. By comparing the gene expression profiles of NB and sham tissues, we detected DEMs and DECs that could act as important regulators in developing NB and bladder fibrosis. Additionally, a co-expression network (circRNA–miRNA–mRNA) was established to elucidate the internal regulatory connections between the DEMs and DECs and demonstrate the ceRNAs regulatory relationships that underlie them.

Materials and Methods

Experimental animal and SSCI model

Twenty 10–week–old female Wistar rats weighing 200 ± 20 g (Beijing Charles River Laboratories Animal Technology Co., Ltd., Beijing, China) were numbered 1–20 and divided into 4 groups according to a random number table (N =5/group), including the sham group and the SSCI [1–3] groups (1W–, 2W–, and 4W–post–SSCI). All animals were housed in separate cages under well-ventilated conditions with a circadian rhythm of 12/12 h, room temperature of 24 to 26 °C, humidity of 50% to 70%, and free access to water and food. After 1 week of adaptive feeding, all rats were fasted for 24 h and anesthetized with isoflurane inhalation (3%) to reduce the pain or distress of the rats. Rats in the SSCI groups were subjected to Hassan Shaker spinal cord transection (Shaker et al., 2003) to construct the SSCI model. The Institutional Animal Care and Use Committee of Xuanwu Hospital Capital Medical University granted approval for all experimental procedures(No. XW-20210910-1), which were conducted in accordance with the National Institute of Health Guidelines for the Care and Use of Laboratory Animals. The schematic representation of the experimental design was shown in Fig. 1.

Figure 1 Schematic diagram of the experimental design.

The injured segment of this model was uniformly fixed at the T10–T11 segment (Inskip et al., 2009). The method was as follows: The lowest group of floating ribs connected to the 13th thoracic vertebra was used as bony landmarks to locate. Skin preparation and disinfection were performed after determining the T10–T11 level. Longitudinal incision of approximately 2 to three cm was then centered on the vertebral body along the long axis of the spine. The skin and subcutaneous fascia were incised sequentially, and the erector spinae muscles on both sides were bluntly separated to expose the spinous and paraspinal muscles. Muscles were bluntly separated from their attached spinous processes with curved forceps to expose further the spinous processes of T10 and T11 and the adjacent pedicles. The spinal cord was exposed by removing the T10 lamina from the caudal to the cranial side with a microrongeur until the pedicles were on both sides. The spinal cord was hooked out of the intervertebral space with a dental probe and quickly cut off along the hook with a scalpel tip. This can be repeated to ensure complete spinal cord transection. Finally, the two broken ends of the spine were gently lifted with curved forceps to ensure complete spinal cord transection. Injury to the spinal cord capsule was avoided throughout. After observing the heartbeat and respiration of rats without abnormality, the muscles, fascia, and skin were sutured in succession. The incision and surrounding area were disinfected with 5% complexed iodine, and 2 × 105 U of penicillin sodium was injected intraperitoneally. The sham group underwent the same surgical preparation as the experimental group and received laminectomy at the same level. All rats were single-housed after surgery. Clean and dry bedding was changed daily. Reduce the height of feed and water appropriately to ensure that rats can eat independently. Artificial assisted voiding was performed every 8 h in the conscious state of rats using the Crede maneuver until the bladder regained the micturition reflex. Penicillin sodium 2 × 105 U was injected intraperitoneally daily for 5 days after surgery to prevent infection.

Inclusion and exclusion criteria

Inclusion criteria: 1. Hind limb motor function: rats’ hind limbs in the experimental group were observed if they participated in walking. If the hindlimb was only dragged while walking on the forelimb and could not participate in walking, the Basso, Beattie, and Bresnahan (BBB) score for SSCI in rats was 0. It was then considered a successful SSCI model. 2. Bladder voiding function: In the surgical anesthesia and spinal shock period, the bladder was observed to see if the state of retention and bladder distension were obvious, and spontaneous urination could not be performed. However, the bladder capacity decreased after the spinal shock period, and its distension was not obvious. No manipulation was required to assist urination, and the lower abdomen and bedding material in the cage were humid, indicating that the detrusor muscle of the bladder in rats had produced uninhibited contraction.

Exclusion criteria: Model rats were not included in the experiment if they showed the voluntary movement of both hind limbs, spontaneous urination, autophagy, or death.

Bladder tissue collection

For ethical reasons, all animals were euthanized according to AVMA guidelines. We administered intraperitoneal injections of pentobarbital sodium at a dosage of 150 mg/kg to perform euthanasia, thereby reducing the pain or distress of the rats. Bladder tissues were collected at 1, 2, and 4 weeks after modeling in the SSCI [1–3] groups, respectively, and at 4 weeks after surgery in the sham group. Each tissue was divided into three equal fractions. One-third of the samples were preserved in 4% paraformaldehyde, embedded in paraffin, and then sectioned for morphological observation using hematoxylin-eosin (HE) and Masson staining. Another one-third was used to extract total RNA for transcriptome high-throughput sequencing. The last one-third was used for quantitative real-time polymerase chain reaction (qRT-PCR) validation.

HE staining and Masson staining

Pathological analyses were performed to assess the size of smooth muscle cells in the bladder and the presence of bladder fibrosis. The bladder tissues were fixed overnight with a 4% paraformaldehyde solution, then dehydrated in a series of ethanol and cleared in xylene. The tissues were subsequently embedded in paraffin and cut into sections with a thickness of 6 micrometers. HE staining detected the nuclei, bladder smooth muscle cells, and bladder morphology. Masson staining showed collagen fibers (blue) and muscle tissue (red) to detect the severity of bladder fibrosis. The staining signals were quantitatively analyzed using Image J software (version 1.53q).

RNA isolation and cDNA library construction and Illumina sequencing

Three bladders were obtained from each group for whole transcriptome sequencing. TRIzol reagent (Invitrogen, Carlsbad, CA, USA) was used to extract total RNA from each tissue. RNA concentration in each tissue was measured using NanoDrop ND-2000 (NanoDrop, Wilmington, DE, USA). The RNA Integrity Number (RIN) was determined using the Agilent 2100 Bioanalyzer (Applied Biosystems, Carlsbad, CA, USA); all tissues had RIN ≥7.0. Twelve cDNA libraries were generated, each containing three different tissue samples to analyze the expression patterns of both mRNAs and circRNAs. According to the protocol, ribosomal RNAs (rRNAs) were depleted using TruSeq Small RNA Sample Prep Kits (Illumina, San Diego, CA, USA). Afterward, the TruSeq Stranded Total RNA Library Prep Kit (Illumina, San Diego, CA, USA) was utilized to construct the cDNA libraries. Quantification of libraries was determined using the Agilent 2100 Bioanalyzer (Applied Biosystems, Carlsbad, CA, USA). Each group had 3 biological replicates when performing RNA sequencing (RNA-seq) at approximately 72X sequencing depth. The statistical power of this experiment was 0.86 as calculated in RNASeqPower (https://doi.org/doi:10.18129/B9.bioc.RNASeqPower). The Illumina Hiseq 4000 Sequencer (Illumina, Shanghai, China) was used to conduct paired-end reads mode sequencing of 150 bp on the final cDNA libraries.

Discovery of the mRNAs and circRNAs

The raw reads underwent quality control using Trimmomatic software (version 0.36) with the default quality threshold to remove the containing adaptors, excessive N content of unknown bases, and low-quality tags. After this step, the clean reads were ready for mapping and subsequent analysis. FastQC (version 0.11.5) was used to establish data quality sequence that persisted after filtration. High-quality clean reads were mapped to the reference genome Rnor_6.0 by Hisat2 software (version 2.0.5) with the rna-strandness parameter. The mapped reads from each library were assembled using StringTie2 software (version 1.3.3b), and a unified transcriptome was created by combining all the tissues. CIRI2 software (version 2.0.3) and circAtlas (version 2.0) were used to identify the circRNAs.

Functional analysis of DEMs and DECs

The readings of the SSCI [1–3] groups were individually compared to the sham group. DESeq2 software (version 1.18.0) was used, which employs the negative binomial distribution to determine the differential expression between the SSCI and sham tissues. Volcano plots and hierarchical clustering were used to characterize and identify the DEMs and DECs by R software (version 3.22.3) with absolute fold change (FC) >2 (p < 0.05). The logarithmic transformation was applied to the FC values of the matched groups. A significance level of p < 0.05 was used as the threshold for identifying DEMs and DECs. The RNA expression of the sham group was used as the denominator during FC computation. The DEMs and DECs with log2 (FC) <−1 were considered down-regulated, while those with log2 (FC) >1 were considered up-regulated. The DEMs and DECs associated with each NB stage were subjected to Venn diagram analysis. Thus, we determined the commonly expressed DEMs and DECs that potentially up-regulated or down-regulated gene expression in the three-time points of NB.

The Miranda software (version 3.3a) predicted the DECs-targeted miRNAs to determine the functions of DECs in bladder fibrosis and NB following SSCI. The target genes of DECs were analyzed using R software (package clusterProfiler) to perform Gene Ontology (GO) and Kyoto Encyclopedia of Genes and Genomes (KEGG) analyses. The target miRNAs of DECs and the target mRNAs of miRNAs were predicted using TargetScan, MiRanda, and RNAhybrid. A Cytoscape-based approach (http://cytoscape.org/) was used to construct the circRNA–miRNA–mRNA interaction network.

qRT-PCR Validation

The top 10 co-expressed DECs in each NB stage were subjected to qRT-PCR to verify their expression. TRIzol reagent was used to extract total RNA from bladder tissues. After RNA extraction, rRNAs were depleted using TruSeq Small RNA Sample Prep Kits (Illumina, San Diego, USA). Subsequently, the RNA samples were fragmented and reverse transcribed into cDNA using random primers via the PrimeScript RT reagent kit(Takara Bio Inc., Japan). Then, qRT-PCR was performed using TB Green Premix Ex Taq II (Takara Bio Inc., Shiga, Japan). Gapdh served as the internal control for normalization. All primers for qRT-PCR were synthesized by Tianyi Huiyuan Co. (Beijing, China).

Statistical analysis

Statistical analyses were performed using SPSS software (version 24.0) and GraphPad Prism (version 8.0). The mean ± standard error of mean(SEM) was used to present normally distributed data. The t-test was utilized to determine significant differences in circRNAs and mRNAs expressions between groups, and statistical significance was set at a p-value of <0.05.

Results

Validation of NB following SSCI model in rats

HE and Masson staining (Figs. 2A–2H) were used to verify whether the SSCI rat model was successfully established and evaluate the bladder fibrosis degree. Furthermore, we conducted a detailed quantitative analysis of the staining signals using Image J software (Fig. 2I). The results showed that only a small amount of collagen fibers was seen in the muscular layers of the bladder in the sham group. In the SSCI [1–3] groups, the normal fibrous connective tissue in the bladder wall gradually disordered; the layered structure of the bladder wall gradually disintegrated, and the bladder wall first thinned and then thickened with the time of injury; the lamina propria decreased, smooth muscle hypertrophy and thickening; intermuscular collagen fibers gradually increased, and the staining gradually deepened. According to all the above features, NB following the SSCI model had been established successfully.

Figure 2 HE staining (×50) and Masson (×50) staining in rat bladder tissue.

(A) HE staining of bladder tissues in the sham group. (B) Masson staining of bladder tissues in the sham group. (C) HE staining of bladder tissues in the SSCI–1 group. (D) Masson staining of bladder tissues in the SSCI–1 group. (E) HE staining of bladder tissues in the SSCI–2 group. (F) Masson staining of bladder tissues in the SSCI–2 group. (G) HE staining of bladder tissues in the SSCI–3 group. (H) Masson staining of bladder tissues in the SSCI–3 group. (I) The quantitative analysis of collagen area in Masson staining. Error bars represent mean ± SEM. *: p-value <0.05 vs. sham group, **: p-value <0.01 vs. sham group, ***: p-value <0.001 vs. sham group, ****: p-value <0.0001 vs. sham group.

Results of sequencing and characteristics of transcripts

Using the Illumina platform, 12 cDNA libraries have been sequenced, resulting in approximately 187 Gb of sequenced data. The clean reads of each tissue were compared with the Rnor_6.0 reference genome, and the alignment rate ranged from 93.54 to 94.76% (Table S1). The valid ratio for each library is displayed in Table S2. An examination was conducted on mRNAs and circRNAs expression profiles in both the sham and SSCI [1–3] groups. The results showed that 22,601 mRNAs (Fig. S3) and 7,250 circRNAs (Fig. S4) were identified in all bladder tissues.

Figure 3 MRNAs expression profiles of SSCI 1–3 groups compared to sham group.

(A) Volcano plots compare the DEMs in the SSCI–1 group vs. the sham group. (B) Volcano plots compare the DEMs in the SSCI–2 group vs. the sham group. (C) Volcano plots compare the DEMs in the SSCI–3 group vs. the sham group. mRNA levels are indicated in the plots by red dots for upregulated genes, green dots for down-regulated genes, and gray dots for genes that show no significant changes. (D) Bar diagrams show the number of up-regulated and down-regulated DEMs between the SSCI 1–3 groups and sham group. (E) Hierarchical clustering analysis of DEMs between SSCI 1–3 groups and sham group (FC > 2; p < 0.05). (F) Venn diagrams show the unique and shared sets of the up-regulated DEMs in the SSCI 1–3 groups compared to the sham group. (G) Venn diagrams show the unique and shared sets of the down-regulated DEMs in the SSCI 1–3 groups compared to the sham group.

Figure 4 Expression profiles of circRNAs in SSCI 1–3 groups compared to sham group.

(A) Volcano plots compare the DECs in the SSCI–1 group vs. the sham group. (B) Volcano plots compare the DECs in the SSCI–2 group vs. the sham group. (C) Volcano plots compare the DECs in the SSCI–3 group vs. the sham group. circRNA levels are indicated in the plots by red dots for upregulated genes, green dots for down-regulated genes, and gray dots for genes that show no significant changes. (D) Bar diagrams show the number of up-regulated and down-regulated DECs between the SSCI 1–3 groups and the sham group. (E) Hierarchical clustering analysis of DECs between SSCI 1–3 groups and sham group (FC > 2; p < 0.05). (F) Venn diagrams show the unique and shared sets of the up-regulated DECs in the SSCI 1–3 groups compared to sham group. (G) Venn diagrams show the unique and shared sets of the down-regulated DECs in the SSCI 1–3 groups compared to sham group. (H) Bar diagrams show the distribution of DECs host genes in rat chromosomes. (I) Bar diagrams show the length distribution of the DECs.

The mRNAs expression profiles in sham group and SSCI [1–3] groups

Volcano maps were used to assess the locations of differential mRNAs of the SSCI [1–3] groups vs. the sham group (Figs. 3A–3C). Compared with the sham group, 3,255 (1,885 up-regulated and 1,370 down-regulated), 3449(2052 up-regulated and 1,397 down-regulated), and 884 (254 up-regulated and 630 down-regulated) DEMs were detected in the SSCI–1, SSCI–2, and SSCI–3 groups, respectively. The SSCI [2–3] groups also exhibited some DEMs compared to the SSCI–1 group. Specifically, there were 1,056 and 3,228 DEMs in the SSCI–2 and SSCI–3 groups, with 636 and 1,077 DEMs showing up-regulation, and 420 and 2151 DEMs showing down-regulation, respectively. Compared with the SSCI–2 group, the SSCI-3 group had 3,024 DEMs, with 937 showing up-regulation and 2,087 showing down-regulation (Fig. 3D). Clustered heatmap showed differences in mRNAs expression patterns between the SSCI [1–3] groups and the sham group (Fig. 3E). Venn diagrams of the up-regulated and down-regulated DEMs showed that 420 (115 up-regulated, 305 down-regulated) coexisted in the intersections of the SSCI [1–3] groups vs. the sham group (Figs. 3F–3G). The specific comparisons of DEMs between each group could be found in Fig. S3.

The circRNAs expression profiles in sham group and SSCI [1–3] groups

Volcano maps were used to assess differential circRNAs locations of the SSCI [1–3] groups vs. the sham group (Figs. 4A–4C). The circRNAs expression profiling data showed that 1,339, 1,324, and 1,151 DECs were differentially expressed between the SSCI [1–3] groups and the sham group, of which 408, 349, and 532 were up-regulated, and 931, 975, and 619 were down-regulated, respectively. The SSCI [2–3] groups also exhibited some DECs compared to the SSCI–1 group. Specifically, there were 746 and 1,135 DEMs in the SSCI–2 and SSCI–3 groups, with 332 and 751 DECs showing up-regulation, and 414 and 384 DECs showing down-regulation, respectively. Compared with the SSCI–2 group, the SSCI–3 group had 1,120 DECs, of which 816 were up-regulated, and 304 were down-regulated (Fig. 4D). According to the clustered heatmap, circRNAs expression patterns varied between the SSCI [1–3] groups and the sham group (Fig. 4E). Venn diagrams of the up-regulated and down-regulated DECs showed that 412 (52 up-regulated, 360 down-regulated) coexisted in the SSCI [1–3] groups vs. the sham group intersections (Figs. 4F–4G). Additional analyses of the DECs were conducted to investigate the molecular features of circRNAs, focusing on their length and chromosomal distribution. Fig. 4H revealed that DECs majority were derived from chromosome 1 (10.83%; 785/7250), followed by chromosome 2 (8.0%; 580/7250), and chromosome 5 (6.94%; 503/7250), in descending order. Furthermore, Fig. 4I demonstrated that a significant proportion of the DECs had lengths of less than 2,000 nucleotides (85.94%; 6233/7250). Tables 1 and 2 presented the top 10 up-regulated and down-regulated DECs, respectively, that are co-expressed in all SSCI [1–3] groups when compared with the sham group. The specific comparisons of DECs between each group could be found in Fig. S4.

Table 1 The top 10 up-regulated DECs co-expressed in all SSCI 1–3 groups compared with the sham group.

	CircRNA ID	Fold change	Gene ID	Length	Host gene	
circRNA 3621	Chr2:184309074-184309661−	8.93/7.90/7.94	100360914	588	Fbxw7	
circRNA 5775	Chr6:108797786-108802113 +	8.73/8.48/7.18	299198	294	Fcf1	
circRNA 0617	Chr1:243014234-243018525 +	8.67/8.88/6.92	499337	248	Dock8	
circRNA 2378	Chr15:70098761-70139170−	8.67/8.29/6.66	290396	660	Diaph3	
circRNA 0613	Chr1:242716557-242737019−	8.64/8.68/8.21	679990	782	Pgm5	
circRNA 2146	Chr14:88724076-88725530−	8.18/6.92/7.80	360980	360	Tns3	
circRNA 4403	Chr3:145092634-145129362 +	8.03/8.06/7.22	362235	782	Syndig1	
circRNA 1266	Chr10:109754888-109755050 +	7.95/7.93/8.09	360678	163	Arhgdia	
circRNA 0544	Chr1:220039308-220042767−	7.88/7.16/7.40	293663	1533	Rbm4	
circRNA 0722	Chr1:261165387-261165588−	7.81/7.75/7.60	24642	202	Pgam1	

Table 2 The top 10 down-regulated DECs co-expressed in all SSCI 1–3 groups compared with the sham group.

	CircRNA ID	Fold change	Gene ID	Length	Host gene	
circRNA 3111	Chr19:15151107-15207439 +	−8.15/−8.12/−8.59	113902	44991	Ces1d	
circRNA 0586	Chr1:231457046-231498197−	−7.62/−7.59/−8.05	25565	360	Tle4	
circRNA 1617	Chr12:31203176-31238119−	−7.3/−7.28/−7.74	689257	1457	Adgrd1	
circRNA 4426	Chr3:150692740-150696569−	−7.31/−7.28/−7.75	311567	405	Itch	
circRNA 5664	Chr6:71078256-71089829−	−6.94/−6.91/−7.38	85421	1209	Prkd1	
circRNA 0143	Chr1:80051932-80052296−	−6.94/−6.91/−7.38	292687	253	Qpctl	
circRNA 1498	Chr12:4448685-4450137−	−6.93/−6.91/−7.37	304208	292	Cers4	
circRNA 3098	Chr19:10788035-10789787−	−6.8/−6.78/−7.24	689249	248	Rspry1	
circRNA 4664	Chr4:64451679-64474930−	−6.77/−6.75/−7.21	688705	481	Dgki	
circRNA 4884	Chr4:152937981-152938845 +	−6.77/−6.75/−7.21	312678	307	Kdm5a	

Validation of the expression of circRNAs by qRT-PCR

The RNA-seq analysis showed that circRNA3621, circRNA5775, circRNA0617, circRNA2378, and circRNA0613 were the five most significantly up-regulated DECs common to SSCI [1–3] groups compared with the sham group. Moreover, circRNA3111, circRNA0586, circRNA1617, circRNA4426, and circRNA5665 were the five most significantly down-regulated DECs common to SSCI [1–3] groups compared with the sham group. The above 10 circRNAs were selected for qRT-PCR to confirm the RNA-seq results’ accuracy, and the sequences were listed in Table 3. The qRT-PCR experiment results, presented in Fig. 5, were consistent with the RNA-seq data, thus affirming the RNA-seq results precisions.

Table 3 Primer sequences for qRT-PCR.

Name	Primer sequences	
circRNA 3621	Forward	ACCAAGACAACAAAGAATGTGAA	
Reverse	CAGAGAGCCTCCAGTTCGTC	
circRNA 5775	Forward	ACTGTCTGTACGCCAAGTGT	
Reverse	TTCTTTACTTAGTGCCACACGA	
circRNA 0617	Forward	TTGTGAACCGGAACCTCAGC	
Reverse	AAGTGCGTCATCACAAGTCCT	
circRNA 2378	Forward	TCAGAACTTCATTTATGATGAGCAA	
Reverse	AGCTTAAAGAGAATAGCAGTGAGC	
circRNA 0613	Forward	GGACAGATTGGTCGGCTGATT	
Reverse	CCACCAGCTGCTTTGATCTTC	
circRNA 3111	Forward	CAGCACAGGGGATGAACACA	
Reverse	GACACTTTCACCTCCTGCTGA	
circRNA 0586	Forward	AGAGACTCCATCAAGGCAGAGA	
Reverse	GGTGCTCTTGGGACAGGAAA	
circRNA 1617	Forward	AACTGGGTTTGAGGTCTTAGCC	
Reverse	GCACAGTGAGGTTGAGGGTT	
circRNA 4426	Forward	TCAGAGGTCATCTCAGCCAAAC	
Reverse	TGACTGCCCATCTACTGTGA	
circRNA 5664	Forward	TACAGCAAAAGTCTCCATCTGAG	
Reverse	GTGTGTGGCACCTTCACCTT	
GADPH	Forward	ACAAGATGGTGAAGGTCGGTG	
Reverse	AGAAGGCAGCCCTGGTAACC	

Figure 5 Confirmation of the expression of the DECs using qRT-PCR.

Error bars represent mean ± SEM. *: p-value <0.05 vs. sham group, **: p-value <0.01 vs. sham group, ***: p-value <0.001 vs. sham group, ****: p-value <0.0001 vs. sham group.

The feature and the potential function of DEMs and DECs

GO enrichment analysis was performed, during which biological process (BP), cellular component (CC), and molecular function (MF) were described to filter the key regulators and pathways of the DEMs and the hosting genes of the DECs in the process of NB following SSCI.

For DEMs, the SSCI [1–3] groups were compared to the sham group (Fig. S6); the top 10 up-regulated and the top 10 down-regulated GO terms of DEMs were shown in Figs. 6A–6F. In the up-regulated GO terms, the BP category was mostly enriched in “inflammatory response”, “inflammatory response”, and “chronic inflammatory response”. The CC category was mostly enriched in “extracellular space”, “extracellular space”, and “extracellular space”; the MF category was mainly involved in “integrin binding”, “cytokine activity”, and “serine-type endopeptidase activity”. In the down-regulated GO terms, the BP category was mostly enriched in “potassium ion transmembrane transport”, “potassium ion transport”, and “positive regulation of synapse assembly”; the CC category was mostly enriched in “voltage-gated potassium channel complex”, “integral component of the plasma membrane”, and “anchored component of membrane”; the MF category was mostly involved in “calcium ion binding”, “calcium ion binding”, and “calcium ion binding”. In addition, comparing the SSCI [1–3] groups to the sham group (Fig. S7), the top 20 up-regulated and down-regulated KEGG pathways are shown in Figs. 7A–7F. Among those, the most significantly up-regulated pathways were “Cytokine-cytokine receptor interaction”, “Cytokine-cytokine receptor interaction”, and “Cytokine-cytokine receptor interaction”, respectively. The most obvious down-regulated pathways were the “Neuroactive ligand–receptor interaction”, “Calcium signaling pathway”, and “Neuroactive ligand–receptor interaction”, respectively.

Figure 6 GO enrichment analysis of the DEMs in SSCI 1–3 groups compared to sham group.

(A) Top 10 GO enrichment terms in BP, CC, and MF of the up-regulated DEMs in SSCI–1 group vs. sham group. (B) Top 10 GO enrichment terms in BP, CC, and MF of up-regulated DEMs in SSCI–2 group vs. sham group. (C) Top 10 GO enrichment terms in BP, CC, and MF of up-regulated DEMs in SSCI–3 group vs. sham group. (D) Top 10 GO enrichment terms in BP, CC, and MF of the down-regulated DEMs in SSCI–1 group vs. sham group. (E) Top 10 GO enrichment terms in BP, CC, and MF of down-regulated DEMs in SSCI–2 group vs. sham group. (F) Top 10 GO enrichment terms in BP, CC, and MF of down-regulated DEMs in SSCI–3 group vs. sham group.

Figure 7 KEGG pathway analysis of DEMs in SSCI 1–3 groups compared to sham group.

(A) Top 20 KEGG pathway analysis of up-regulated DEMs in SSCI–1 group vs. sham group. (B) Top 20 KEGG pathway analysis of up-regulated DEMs in SSCI–2 group vs. sham group. (C) Top 20 KEGG pathway analysis of up-regulated DEMs in SSCI–3 group vs. sham group. (D) Top 20 KEGG pathway analysis of down-regulated DEMs in SSCI–1 group vs. sham group. (E) Top 20 KEGG pathway analysis of down-regulated DEMs in SSCI–2 group vs. sham group. (F) Top 20 KEGG pathway analysis of down-regulated DEMs in SSCI–3 group vs. sham group.

The top 30 up-regulated and the top 30 down-regulated GO terms of DECs are shown in Figs. 8A–8F. According to GO analysis (Fig. S8), the terms that contained the most DECs were “negative regulation of neuron migration”, “negative regulation of cell migration”, and “negative regulation of receptor biosynthetic process”, respectively, in the up-regulated BP category. The terms that contained the most DECs were “extracellular region”, “trans-Golgi network membrane”, and “postsynaptic density”, respectively, for up-regulated CC category. For up-regulated MF, the terms that contained the most DECs were “iron ion binding”, “pre-mRNA intronic binding”, and “RNA polymerase II transcription factor activity, sequence-specific transcription regulatory region DNA binding”, respectively. In the down-regulated GO terms, the BP category was mostly enriched in “adenylate cyclase-activating G-protein coupled receptor signaling pathway”, “cAMP biosynthetic process”, and “intracellular signal transduction”; the CC category was mostly enriched in “endocytic vesicle”, “guanylate cyclase complex, soluble”, and “sarcolemma”; the MF category was mostly involved in “calcium- and calmodulin-responsive adenylate cyclase activity”, “calcium- and calmodulin-responsive adenylate cyclase activity”, and “calcium- and calmodulin-responsive adenylate cyclase activity”. In addition, comparing the SSCI [1–3] groups to the sham group (Fig. S9), the top 20 up-regulated and the top 20 down-regulated KEGG pathways are shown in Figs. 9A–9F. Among those, the most significantly up-regulated pathways were involved in “Vasopressin-regulated water reabsorption”, “Notch signaling pathway”, and “Glutamatergic synapse”, respectively. The most obvious down-regulated pathways were involved in the “Phospholipase D signaling pathway”, “Phospholipase D signaling pathway”, and “Bile secretion”, respectively. Go and KEGG analysis indicated that an increasing quantity of cells with immune function compared to the sham group, and the Notch signaling pathway might be the main cause of NB following SSCI.

Figure 8 GO enrichment analysis of the DECs in SSCI 1–3 groups compared to sham group.

(A) Top 10 GO enrichment terms in BP, CC, and MF of the up-regulated DECs in SSCI–1 group vs. sham group. (B) Top 10 GO enrichment terms in BP, CC, and MF of up-regulated DECs in SSCI–2 group vs. sham group. (C) Top 10 GO enrichment terms in BP, CC, and MF of up-regulated DECs in SSCI–3 group vs. sham group. (D) Top 10 GO enrichment terms in BP, CC, and MF of the down-regulated DECs in SSCI–1 group vs. sham group. (E) Top 10 GO enrichment terms in BP, CC, and MF of down-regulated DECs in SSCI–2 group vs. sham group. (F) Top 10 GO enrichment terms in BP, CC, and MF of down-regulated DECs in SSCI–3 group vs. sham group.

Figure 9 KEGG pathway analysis of the DECs in SSCI 1–3 groups compared to sham group.

(A) Top 20 KEGG pathway analysis of up-regulated DECs in SSCI–1 group vs. sham group. (B) Top 20 KEGG pathway analysis of up-regulated DECs in SSCI–2 group vs. sham group. (C) Top 20 KEGG pathway analysis of up-regulated DECs in SSCI–3 group vs. sham group. (D) Top 20 KEGG pathway analysis of down-regulated DECs in SSCI–1 group vs. sham group. (E) Top 20 KEGG pathway analysis of down-regulated DECs in SSCI–2 group vs. sham group. (F) Top 20 KEGG pathway analysis of down-regulated DECs in SSCI–3 group vs. sham group.

The circRNA–miRNA–mRNA network construction

Based on comparing the SSCI groups and the sham group at three different time points, we constructed the circRNA–miRNA–mRNA co-expression network to reveal the complex interactions among circRNAs, miRNAs, and mRNAs. Those DECs predicted miRNAs interacting with them, and the ceRNAs network was calculated and visualized according to those miRNAs (Fig. S10). These circRNAs and their neighbors were found to form a complex module when they were mapped onto a co-expression network. As shown in Fig. 10, circRNA1249 and circRNA2239 were the network’s largest node, indicating that the circRNA1249-mediated and circRNA2239-mediated network may play a crucial role in the NB process following SSCI.

Figure 10 Network depicting the co-expression of circRNAs, miRNAs, and mRNAs.

(A) CircRNA–miRNA–mRNA co-expression network in the SSCI–1 group vs. the sham group. (B) CircRNA–miRNA–mRNA co-expression network in the SSCI–2 group vs. the sham group. (C) CircRNA–miRNA–mRNA co-expression network in the SSCI–3 group vs. the sham group. CircRNAs are highlighted with yellow nodes, while the target miRNAs and mRNAs are represented by green and red nodes, respectively.

Discussion

NB following SSCI is an interstitial bladder disease with the structural remodeling of bladder tissue and matrix over-deposition (Comperat et al., 2006). Following SSCI, a series of pathological changes developed in the bladder, including irreversible atrophy, extracellular matrix formation, and connective tissue accumulation (Bushnell et al., 2022). Irreversible atrophic fibrosis of the detrusor muscle after SSCI is an important cause of poor recovery of bladder function (Hamid et al., 2018). There is a lack of proven therapeutics for NB following SSCI and bladder fibrosis yet. Therefore, it is crucial to comprehensively comprehend the molecular mechanisms that occur in NB following SSCI. In the present study, tissue staining showed that detrusor fibrosis was not evident in the bladder at the early stage of SSCI. However, fibrosis gradually worsened with the extension of SSCI time. After 2 weeks of SSCI, the cross-sectional area of collagen fibers gradually increased, which was most obvious at 4 weeks after SSCI. This result is consistent with previous reports (Liu et al., 2016). It also proved that the NB following the SSCI rat model had been successfully established in this study. Preventing bladder fibrosis is critical during NB repair after SSCI, but the specific mechanism of fibrosis remains unknown.

The miRNAs do not code for proteins but is essential in regulating gene expression. It is vital in many essential processes, including development, normal physiological functions, and certain diseases such as bladder fibrosis and NB following SSCI (Shang et al., 2020). Our previous studies showed that miR-21-5p may promote TGF- β1 expression and enhance the fibrotic response. Furthermore, some fibrosis-related genes, such as smad7, could be down-regulated by miR-21-5p (Cui et al., 2019). We also observed that lncRNAs could play a role in regulating NB following SSCI and bladder fibrosis. Our previous studies showed that some lncRNAs might affect the bladder fibrosis process following SSCI through certain signaling pathways (Ruan et al., 2022). For instance, in bladder fibrosis following SSCI, the lncRNA Mir155hg targets miR-155 and activates the fibrotic signaling pathway downstream, including the TGF- β1 pathways. Furthermore, lncRNA H19 could promote bladder fibrosis by regulating epithelial-mesenchymal transition (EMT).

At present, circRNAs have been proven to be closely related to nerve and muscle function. CircRNAs regulated axonal regeneration and nerve repair during the repair pathophysiology after peripheral nerve injury (Sohn & Park, 2020). Circ zfp609, circ LMO7, and circ FUT10 played key roles in maintaining normal muscle function and regulating myogenesis, proliferation, and differentiation by sponging miRNAs (Li et al., 2018a; Wang et al., 2019b; Wei et al., 2017). Studies have shown that circRNAs can be involved in fibrosis in multiple organs such as the liver, heart, lung, and bladder. For example, circ 0071410 was associated with radiation-induced liver fibrosis. Inhibition of circ 0071410 expression upregulated miR-9-5p and attenuated radiation-induced activation of hepatic stellate cells (Chen et al., 2017). Circ ZC3H4 and its downstream product ZC3H4 were involved in silica-induced lung macrophage activation, and activated macrophages could promote lung fibroblast proliferation and migration through the circ ZC3H4/ZC3H4 pathway (Yang et al., 2018b). Circ PVT1 was a inhibitor of miR-203, promoting bladder outlet obstruction-induced bladder hypertrophy and fibrogenesis (Li et al., 2022).

Based on the above studies, circRNAs may be an important regulator of SSCI-induced NB and bladder fibrosis. Then we used transcriptome high-throughput sequencing to identify the possible mechanisms for NB following SSCI and bladder fibrosis through DEMs and DECs. Several genes and enriched terms were associated with the immune, nervous, and urinary systems. Sequencing results revealed 7,250 circRNAs in three SSCI and sham bladder tissue, which accounts for approximately 7.22% of the total known circRNAs. There were differentially expressed up-regulated and down-regulated circRNAs in the three-time points after SSCI compared with the sham group. Among them were 2,332 DECs, including 1,013 up-regulated and 1,319 down-regulated circRNAs. In the SSCI–1 group, 408 DECs were up-regulated, and 931 were down-regulated; in the SSCI–2 group, 349 were up-regulated, and 975 were down-regulated. However, 532 were up-regulated and 619 were down-regulated in the SSCI–3 group. The above results showed many DECs in SSCI induced NB model, which preliminarily suggested that some DECs may play an important role in SSCI-induced NB and bladder fibrosis. It provides new insights into the mechanistic role of circRNAs in SSCI-induced NB, which is the first study to assess their expression patterns.

The top five most significantly up-regulated DECs (circRNA3621, circRNA5775, circRNA0617, circRNA2378, and circRNA0613) and the top five most down-regulated DECs (circRNA3111, circRNA0586, circRNA1617, circRNA4426, and circRNA5665) were selected respectively and their expression levels were detected using qRT-PCR to validate our RNA-seq results. The results verified the expression patterns of the above 10 DECs identified by RNA-seq. Due to their role as miRNAs inhibitor, circRNAs may be involved in SSCI-induced NB and bladder fibrosis through their interactions with miRNAs. Among those up-regulated circRNAs, the host gene of circRNA3621 is Fbxw7. It has been reported that Fbxw7 may be associated with organ fibrosis, cell cycle, differentiation, and apoptosis in lung fibrosis-associated cancer (Guyard et al., 2017; Wang et al., 2020). In profibrotic macrophages, Fbxw7 could inhibit TGF- β expression by interacting with c-Jun (He et al., 2021). It was also involved in the transition of EMT through the mTOR pathway (Diaz & De Herreros, 2016). Consistent with these studies, our data showed that circRNA3621 was upregulated 8.93-, 7.90-, and 7.94-fold in SSCI [1–3] groups, respectively.

Furthermore, the host gene Dock8 of circRNA0617 was shown to regulate immune function-associated signal transduction and the shape and integrity of lymphocytes in the previous study (Kearney, Randall & Oliaro, 2017; Su et al., 2019). Our findings showed that circRNA0617 was up-regulated in NB tissues compared with the sham group. Interestingly, the host gene Diaph3 of circRNA2378, which functioned as a positive mediator of the TGF- β pathway promoting EMT progression, was highly expressed in lung, mammary, and renal epithelial cells (Rana et al., 2018). In addition, Pgam1, the host gene of circRNA0722, interacted with TGF and mediated inflammation, apoptosis, and collagen accumulation, further contributing to fibrosis progression (Wu et al., 2021). However, we have previously reported that NB and bladder fibrosis after SSCI is associated with the TGF- β pathway and EMT (Ruan et al., 2022). Our present sequencing data showed upregulation of circRNA2378 and circRNA0722, suggesting they contribute to bladder fibrosis after SSCI.

Among those down-regulated circRNAs, the host gene of circRNA0586 is Tle4. Tle4 was reported to possibly increase the expression level of vimentin, decrease the expression level of E-cadherin, and promote the progression of EMT (Wu et al., 2016). This study also found that circRNA4426 and its host gene Itch were down-regulated and may play a role in organ fibrosis and inflammatory processes. For example, circRNA Itch acted directly on the miR-33a-5p/SIRT6 axis to regulate renal inflammation and fibrosis in diabetic mice (Liu et al., 2021). Moreover, Itch could inhibit intestinal fibrosis due to inflammatory bowel diseases by decreasing IL-17 expression (Paul et al., 2018). Yang et al. (2018a) reported on circRNA4426 in bladder diseases suggested that circ-Itch can trigger cell cycle arrest and apoptosis in bladder cancer, thereby inhibiting the proliferation, invasion, and migration of bladder cancer cells.

Cytoscape software was used to analyze circRNA–miRNA–mRNA interaction networks of DECs and their potential miRNAs targets to better understand the role of circRNAs in SSCI-induced NB. The network revealed rno-circ-2239 (micu2) as the largest node, indicating that the rno-circ-2239–miRNA–mRNA–mediated network may play a critical role in the pathogenesis of SSCI-induced NB. Numerous DECs were present in NB fibrosis due to SSCI. However, little was known about circRNAs expression and function. CircRNAs are shown to be functionally closely related to host genes (Dai et al., 2021). Therefore, this study combined GO analysis, KEGG pathways, and circRNA–miRNA–mRNA co-expression network analysis to predict the possible function of DECs at each time point after SSCI and to understand their functional characteristics and regulatory pathways over time. So far, no research evidence indicates the specific expression patterns of circRNAs and the corresponding pathways that are enriched in NB tissue after SSCI. All three groups of SSCI-associated circRNAs were similarly enriched in GO terms, such as “inflammatory response”, “immune response”, “extracellular space”, and “cytokine secretion”. This indicated that the quantity of immune function-related cells increased during the occurrence and development of SSCI. “Inflammatory response” and “immune response” played crucial roles in this process. A recent study demonstrated that mice exhibited a series of immune-inflammatory reactions in bladder tissue following SCI, including cytokine-mediated signaling and production of interleukin-6. Furthermore, transcriptomic sequencing confirmed the upregulation of genes involved in immune response (CXCL13, SYK, and CCR7) and inflammation (CCR2, CX3CL1, and TREM2) (Von Siebenthal et al., 2023). The KEGG analysis also revealed that circRNAs in NB tissue linked to SSCI were significantly enriched in pathways, such as the “IL-17 signaling pathway”, the “TGF-beta signaling pathway”, the “cAMP signaling pathway”, the “PI3K-Akt signaling pathway”, and the “MAPK signaling pathway”. Evidence demonstrated that certain pathways play a role in developing organ fibrosis. For example, RNA-seq and single-cell sequencing were performed on rat liver tissues with CCl4-induced fibrosis, suggesting that fibrosis is closely related to the Chemokine signaling pathway (Liu et al., 2022b).

Additionally, a study showed that IL-17 could promote myocardial fibrosis via PKC β/Erk1/2/NF- κB signaling pathway (Liu et al., 2012), and the cAMP signaling pathway may play a critical role in idiopathic pulmonary fibrosis (Wang et al., 2021). The PI3K-Akt signaling pathway has been demonstrated to play a role in the process of EMT and the development of fibrosis in various organs, including the kidney, liver, and lung (Wang et al., 2019a; Wu et al., 2017; Xu et al., 2022). Therefore, we suspect this pathway also plays an important role in bladder fibrosis after SSCI. Moreover, MAPK signaling has been shown to be involved in the EMT process to promote renal fibrosis further (Liu et al., 2022a). To summarize, identifying DECs involved in these pathways suggests that their expression changes may be responsible for bladder fibrosis after SSCI. The bioinformatics analysis conducted in this study found significant alterations in several circRNAs and mRNAs expression patterns in the bladder following SSCI. It is premature to consider these DECs as either promising targets for therapeutic intervention or potential biomarkers for NB at this stage.

However, the upregulation of DECs after SSCI is not solely driven by transcriptional enhancement or pathway activation. It may also be influenced by changes in the cellular composition of the tissues. In complex tissues, circRNA expression profiles can vary among different cell types. A comparative analysis of spatial data and bulk tissue data from the same patients revealed that the majority of significantly DECs detected in bulk tissue analysis did not reflect changes occurring within cancer cells. Instead, the observed differential expression was attributed to variations in the proportion of quiescent smooth muscle cells, which exhibit a high content of circRNAs, and fast proliferating cancer cells, which have lower circRNA levels (Garcia-Rodriguez et al., 2023). This finding underscores the importance of investigating circRNA expression at the single-cell level and using spatial transcriptomics techniques in future research to gain a deeper understanding of the cellular heterogeneity and molecular mechanisms underlying changes in gene expression patterns after SSCI.

This study has a few limitations that should be addressed in future research. Firstly, the study did not investigate the detailed interactions between circRNAs and mRNAs in bladder tissue following SSCI. Additional investigation on these DECs as possible biomarkers for NB or primary targets for therapeutic intervention is necessary. Moreover, we intend to investigate the circRNAs profiles in the various strata of the bladder mucosa and detrusor muscles. Finally, because of this study’s limited sample size, further comprehensive prospective research comprising a larger number of rats or human samples is required to confirm these findings. Thus, further exploration of the biological functions and mechanisms of NB following SSCI with respect to circRNAs is necessary.

Conclusions

This study provided a comprehensive view of the expression of circRNAs and mRNAs in NB following SSCI. The results provide strong evidence that circRNAs and mRNAs could potentially be biomarkers for bladder fibrosis and NB following SSCI. It suggested the underlying mechanisms of NB following SSCI from a circRNA perspective. It identified novel circRNAs that may aid in the early detection and management of human NB in the coming years.

Supplemental Information

Table S1 Genome mapping

Click here for additional data file.

Table S2 Summary statistics of sequencing data

Click here for additional data file.

Supplemental Information 3 HE sham

Click here for additional data file.

Supplemental Information 4 HE SSCI–1

Click here for additional data file.

Supplemental Information 5 HE SSCI–2

Click here for additional data file.

Supplemental Information 6 HE SSCI–3

Click here for additional data file.

Supplemental Information 7 Masson sham

Click here for additional data file.

Supplemental Information 8 Masson SSCI–1

Click here for additional data file.

Supplemental Information 9 Masson SSCI–2

Click here for additional data file.

Supplemental Information 10 Masson SSCI–3

Click here for additional data file.

Supplemental Information 11 Gene expression level (FPKM)

Click here for additional data file.

Supplemental Information 12 CircRNAs annotation, CircRNAs length region and chromosome density

Click here for additional data file.

Supplemental Information 13 Raw data for qPCR validation

Click here for additional data file.

Supplemental Information 14 GO enrichment of the DEMs in SSCI 1–3 group vs. sham group

Click here for additional data file.

Supplemental Information 15 KEGG pathway of the DEMs in SSCI 1–3 group vs. sham group

Click here for additional data file.

Supplemental Information 16 GO enrichment of the DECs in SSCI 1–3 group vs. sham group

Click here for additional data file.

Supplemental Information 17 KEGG pathway of the DECs in SSCI 1–3 group vs. sham group

Click here for additional data file.

Supplemental Information 18 CircRNA–miRNA–mRNA co-expression network in the SSCI–1 group vs. the sham group

Click here for additional data file.

Supplemental Information 19 CircRNA–miRNA–mRNA co-expression network in the SSCI–2 group vs. the sham group

Click here for additional data file.

Supplemental Information 20 CircRNA–miRNA–mRNA co-expression network in the SSCI–3 group vs. the sham group

Click here for additional data file.

Supplemental Information 21 RNA sequences data

Click here for additional data file.

Supplemental Information 22 The ARRIVE guidelines 2.0: author checklist

Click here for additional data file.

We are grateful for the experimental support by the laboratory of Xuanwu Hospital Capital Medical University.

Additional Information and Declarations

Competing Interests

Author Contributions

Animal Ethics

Data Availability

The authors declare there are no competing interests.

Jimeng Ruan conceived and designed the experiments, performed the experiments, analyzed the data, prepared figures and/or tables, and approved the final draft.

Xin Cui conceived and designed the experiments, analyzed the data, prepared figures and/or tables, and approved the final draft.

Hao Yan conceived and designed the experiments, performed the experiments, prepared figures and/or tables, and approved the final draft.

Chunsong Jia conceived and designed the experiments, performed the experiments, authored or reviewed drafts of the article, and approved the final draft.

Tongwen Ou conceived and designed the experiments, analyzed the data, authored or reviewed drafts of the article, and approved the final draft.

Zhenhua Shang conceived and designed the experiments, analyzed the data, authored or reviewed drafts of the article, and approved the final draft.

The following information was supplied relating to ethical approvals (i.e., approving body and any reference numbers):

The Institutional Animal Care and Use Committee of Xuanwu Hospital Capital Medical University provided full approval for this research.

The following information was supplied regarding data availability:

The sequences are available at NCBI GEO: GSE227447 and in the Supplemental File.

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
