# Peer review of "Expression profiles of circular RNAs and interaction networks of competing endogenous RNAs in neurogenic bladder of rats following suprasacral spinal cord injury"

_PeerJ, doi:10.7717/peerj.16042_

## Round 0.1 · original submission · Major Revisions

Please address concerns of both reviewers and revise your manuscript accordingly.

·

Basic reporting

80-84 : Writer need to unify the word RNA to the plurals form RNAs when needed through the manuscript. E.g. add an ‘s’ microRNA (miRNA) → microRNAs (miRNAs), lncRNA → lncRNAs …
83 : ncRNAs function as competing endogenous RNAs (ceRNAs) → ncRNAs have several others known and unknown way of function in a cell, it is not only by competition
86-87 : Add reference for the role of CircPVT1 on miR-203v, if it is (Romo et al. 2018) please rewrite line 86 to 89 to merge the common reference.
190 : 500 ng/each tissue is not a useful information just simplify to : “... were depleted using TruSeq Small RNA Sample...”
201: Add the quality threshold used for Trimmomatic software, else indicate it is default parameter
205: Add the mapping parameters used for Hisat2, else indicate it is default parameter
215: simplify to ‘absolute fold change (FC) >2’

217: indicate if you are using the RNA expression of ‘shame’ as denominator during fold change computation so log2 (FC) >1 indicate RNAs that are upregulated in SSCI 1-3 groups

231: how does RNA circularization interfered with Reverse-transcription? Did you add a linearization step?
242: remove word ‘indicating’
315:350: This indicate that some cells with immune function are in increasing quantity vs shame. .c.f discussion
362, 395-400: This information should be in the introduction. For my point of view, and in the context of RNAs, the word ‘inhibitor’ is a better word to use than ‘sponge’
413-503: Like you are using complex tissue, it is not because a RNA is found up-regulated in SSCI that it is a prove of an increase of transcription or an increase in a pathway, this up-regulation can be created by an increase in a cell type that is constitutively expressing this RNA. Add some discussion about the cellular changes and the impact on resulting RNA expression.

Figure 1: You need to orient the reader by adding label in the figure to explain what you are showing in this figure, what is interesting in HE staining vs Masson staining? Where are the collagen fibers? The muscular fibers?...

Figure 9: The names are too small to be readable. Add legend, what means the color green? What means the triangle shape? Square? What are indicating the grey lines?....

Table 1: could go to supplementary material

Experimental design

Okay, standard design. no comment

Validity of the findings

no comment

·

Basic reporting

1. Please provide a schema of the experimental design in Figure 1. The schema would be useful for the reader to understand how the injury was inflicted and the timeline of the bladder tissue collected after suprasacral spinal cord injury.
2. Fig 1: Need proper labelling of the images. The group and type of staining. What are we looking at? Use the arrows to indicate collagen fibers (blue) and muscle tissue (red). The image should be self-explanatory. Please provide a scale bar for each image. Panels E, F, G, and H have different magnifications than panels A, B, C, and D. Please provide a quantitative analysis of the mission staining signal.
3. Fig 4: What does the Y axis represent? What was the control gene?7.
4. Fig 5+7: These figures are very difficult to follow. Too many words in each bar graph. Please find a better way to present these bar graphs. Image resolution should be improved.
5.Fig 6: Each panel should have labels. The image should be self-explanatory. An enrichment score is exactly what?
6. Fig 9: These images are practically unreadable. Please increase image resolution or find a better way to present the data.

Experimental design

1. Why was this study conducted using female rats? A gender-balanced study could have been more useful.
2. The authors should provide H&E and mission staining of a whole neurogenic bladder. Does this condition happen uniformly throughout the bladder or appear in a patchy way at the beginning and then progress? I wonder how that might have an impact on sampling the bladder tissue used for RNAseq (considering 1/3 of the bladder was used).
3. Is there a common set of up- or down-regulated circRNA present in all 3 stages? What is the status of specific up- or down-regulated circRNA or mRNA sets found in stage 1 when compared to stage 2 or stage 3? Do they change?
4. Is there any other molecular marker available for NB? The author should confirm their NB model with more than one marker.

Validity of the findings

1. Line 301: Have you checked one or two circRNAs that didn’t change in expression as a control?
2. Line 417: What is the average number of circRNAs present in tissue? 3090 is what percent of total known circRNA? It would be great to comment on the coverage of RNAseq here.
3. Line 423: How do you know these are causes of NB? Some of the DECs in the early stages could be responsible for developing NB, but during the advanced stage, the DECs could be the results of cellular response due to NB. Is NB reversible? Is it possible to add back a few downregulated circRNAs or ko few upregulated circRNA to test whether it would reverse the condition?
4. Line 447-what is the relevance? Is this circRNA0722 up/down regulated?
5. Is there any available RNAseq data set from Human NB? The authors should check the correlation of their top 10 hits (circRNAs).

Additional comments

The manuscript by Ruan & Cui et al., " Expression profiles of circular RNA and interaction networks of competing endogenous RNA in neurogenic bladder following suprasacral spinal cord injury" describes that a set of differentially expressed mRNA and circular RNA might play roles in bladder fibrosis and neurogenic bladder progression after suprasacral spinal cord injury in rats. The authors used the RNA-seq technique to determine the expression profile of mRNAs and circular RNAs and validated the results by qPCR. The authors have employed two assays to test their hypothesis. While the findings are interesting and could be informative, there are challenges in the basic experimental design, the model used, the way the data is presented, and analyzed. There are still basic questions that need to be addressed.

---

## Round 0.2 · accepted · Accept

All concerns of the reviewers were addressed and the amended manuscript is acceptable now.

·

Basic reporting

No other comments, author's revisions resolved my previous concerns.

Experimental design

No comments.

Validity of the findings

No other comments, author's revisions resolved my previous concerns.

Additional comments

No other comments, author's revisions resolved my previous concerns.

·

Basic reporting

The authors have been responsive to the reviewer's comments and addressed previous concerns in full. They have provided additional supporting data and revised the text for greater clarity, which has strengthened the manuscript considerably.

Experimental design

The authors have been responsive to the reviewer's comments and addressed previous concerns in full. They have provided additional supporting data and revised the text for greater clarity, which has strengthened the manuscript considerably.

Validity of the findings

The authors have been responsive to the reviewer's comments and addressed previous concerns in full. They have provided additional supporting data and revised the text for greater clarity, which has strengthened the manuscript considerably.

Additional comments

The authors have been responsive to the reviewer's comments and addressed previous concerns in full. They have provided additional supporting data and revised the text for greater clarity, which has strengthened the manuscript considerably.